# Association between Grief and Somatic Complaints in Bereaved University and College Students

**DOI:** 10.3390/ijerph191912108

**Published:** 2022-09-24

**Authors:** Lauren Sillis, Laurence Claes, Karl Andriessen

**Affiliations:** 1Faculty of Psychology and Educational Sciences, KU Leuven, 3000 Leuven, Belgium; 2Faculty of Medicine and Health Sciences, University of Antwerp, 2000 Antwerp, Belgium; 3Centre for Mental Health, Melbourne School of Population and Global Health, The University of Melbourne, Melbourne, VIC 3010, Australia

**Keywords:** bereavement, grief, emerging adults, tertiary education, somatic symptoms, physical complaints, pain, social support, intervention

## Abstract

Many emerging adults experience the death of a loved one while they are enrolled as a student in higher education. Bereavement increases the risk of long-term adverse physical and mental health outcomes. Still, as most studies have focused on psychological aspects of grief, little is known about the impact of grief on somatic complaints in students, leaving them vulnerable to health deteriorations. This study aimed to address this gap, and we hypothesized that there is a positive association between grief and somatic complaints in bereaved students. Participants (*N* = 688) were students enrolled at Flemish universities and colleges in Belgium. Participants filled out an online survey with sociodemographic questions, two scales assessing grief, and somatic grief reactions, and two additional questions inquiring whether participants had experienced other somatic reactions, and whether they had taken any steps to remedy their somatic complaints. Regression analyses revealed that less social support, type of relationship (first-degree relative), and the level of grief were positively associated with somatic complaints, and bereaved students reported various complaints such as feeling pain and strains, thus confirming the hypothesis. As bereaved students may be reluctant to seek support for somatic complaints, the findings indicate that information and psychoeducation for bereaved students and their social environment must address somatic grief reactions and encourage timely help seeking. In addition, staff members at psychosocial and medical services for students should be trained to recognize somatic as well as psychological grief reactions. Appropriately skilled, they can inquire about such complaints and provide adequate support to prevent long-term health ramifications.

## 1. Introduction

Emerging adulthood is the transitional stage of life between the ages of 18 and 29, during which emerging adults develop their own identity and an autonomous self [1,2,3,4]. Many emerging adults move away from home, for example, to engage in employment or higher education [5]. While this can be a very gratifying stage of life, due to biopsychosocial developments, they are also vulnerable for developing mental health problems, such as anxiety, mood, and substance abuse disorders, and problems with social functioning [6,7]. 

Many emerging adults experience bereavement while they are enrolled as students in higher education, and between 22% and 30% of students have experienced the death of a close person in the past year [8]. Grief is understood as the primarily affective reaction to a loss, which can be manifested in psychological and somatic reactions [9]. Common grief reactions include, for example, feelings of sadness, anger, guilt, regret, and longing, as well as loss of energy, concentration, and sleep problems [10,11,12]. Bereavement can have a negative impact on students’ academic achievement [13] and development of coping [14]. Bereavement can also increase the risk of a long-term impact on their physical and mental health, for example, regarding posttraumatic stress, and depression, especially after sudden and violent deaths such as suicide [15,16]. 

Studies have reported an association between early parental death and health problems during adulthood, and several somatic complaints have been found in grieving adult populations, such as altered appetite, abnormal weight loss, reduced energy, extreme fatigue, increase or decrease of sexual desire, agitation, and sleep problems [17,18,19]. Furthermore, headache, dizziness, indigestion, chest pain, and muscle weakness [11,19], as well as worsened illness in older bereaved spouses [20], have been reported as possible grief symptoms. 

The literature identified various factors that increase the likelihood of health problems after bereavement. Life-style risk factors include smoking, alcohol and substance use, and a sedentary lifestyle [19,21]. Additionally, a cumulative effect of bereavement distress, socio-economic factors (e.g., academic, work related, or financial issues) and exposure to other distressing events would increase the risk of health problems over the lifespan, indicating a need for both socio-economic and resilience-based interventions to mitigate the long-term impact of bereavement [22]. 

The literature further suggests that physiological processes after bereavement may impact psychological and biological processes and contribute to increased long term vulnerability to stress-related mental and physical disorders [21,22,23]. Short-term stress responses to novel situations, such as bereavement, can be adaptive in regard to behavioural and psychological reactions. However, in the long-term, ongoing physiological stress responses contribute to physical and psychological illnesses such as hypertension, heart disease, infectious illnesses, and depression [21,22,23]. Physiological reactions, especially cortisol dysregulations, have been associated with increased risk of substance use, as well as externalizing and internalizing disorders. 

Research regarding parental loss indicated that a strong relationship with the surviving parent and other family members mitigated the negative physiological effect of parental bereavement. Bereaved emerging adults who experienced strong familial support recovered better from adverse blood pressure and heart stress responses than non-bereaved peers with strong families [24]. The finding suggests that social support and parenting after bereavement contribute to building resilience for future stress [22,24]. While pre-loss factors, such as closeness of the relationship with the deceased person and personal and family history of mental health problems, are associated with grief and mental health problems after the bereavement, a systematic review of the literature has indicated that social support and quality of the relationships after the bereavement attenuate the negative impact of bereavement [25,26].

Some studies have been conducted with bereaved university students. An explorative investigation by Lagrand [27] found that insomnia, headaches, exhaustion, feelings of weakness, digestive disturbances, and nausea were frequently reported by bereaved college students. Other studies emphasized the importance of sleep problems, especially insomnia as a major somatic grief symptom in college students in the first and second year of bereavement [28,29]. Findings from Hall and colleagues [30] indicated that intrusive thoughts of the deceased person interfered with sleep in individuals with bereavement-related depression. Moreover, an increase in the frequency of such thoughts and associated attempts to suppress them led to an increase in physiological arousal, which in turn resulted in longer sleep latency [30]. Herberman Mash and colleagues [4] reported on elevated risks of headache and breathing problems in bereaved students, and Kaiser and colleagues [31] specified that chronic headaches appeared to occur when other grief expressions were impeded. 

Social support is an important source of help for bereaved students [8]. It facilitates self-disclosure, sharing, and processing negative emotions, which enables meaning making and personal growth [32,33,34]. However, many bereaved students are located at a geographical distance from their natural social support network [3,35]. Students may also experience a strong academic pressure, and their culture is primarily oriented at having fun, often with easy access to alcohol and drugs [3,5,36]. Thus, bereaved students can be reluctant when talking about their bereavement, may experience difficulties in finding social support, and peers may lack the skills to adequately listen and provide support [36,37]. 

Bereavement can amplify mental health problems in students, indicating a need to understand grief in this population. However, most studies have focused on psychological grief reactions, and little is known about the impact of grief on somatic complaints in students. A better understanding of somatic grief reactions is crucial to provide adequate support to this vulnerable population and prevent further health deterioration. This study, which is part of a larger research project investigating grief in bereaved students, aimed to address this gap by examining the relation between grief and somatic symptoms. Based on the literature, we hypothesized that there is a positive association between grief and somatic complaints in bereaved students [4,19].

## 2. Materials and Methods

### 2.1. Study Design and Sampling

We recruited students who were between 18 and 28 years old and enrolled in a Flemish college or university in Belgium. Participants had to have experienced the death of a close person between their 12th birthday (to minimize recall bias) and six months before participation in the study. In line with COVID-19-related restrictions, recruitment occurred remotely between March and October 2020. We disseminated the study announcement via approximately 200 student organisations at 18 universities and colleges, and via social media (Facebook, LinkedIn and Twitter). Participants could enrol in a draw to win one of five EUR 20.00 gift vouchers. 

Using G*Power 3.1.9.7 the minimum sample size was estimated at *N* = 143: a linear regression with three independent variables and one dependent variable with *N* = 119 has an effect size of 0.15 with power of 95.10% (α = 0.05). A minimum of *N* = 143 would allow for a 20% of drop-out in participants.

A total of 1390 students started the study questionnaire. Of these, 620 participants were excluded due to incomplete surveys and 79 others who did not fulfil the inclusion criteria (mostly regarding time since bereavement). Finally, three participants with sex ‘other’ were excluded, since this group was too small to include in the analysis. The final sample included *N* = 688 (*M_age_* = 21.39, *SD* = 1.97, range 18–28 years), and 85.6% were female. About 51% of participants had lost a grandparent, 16% a parent, 5% a sibling, 11% another family member, and 17% a friend or other person. Approximately 75% of the deceased persons had died through natural causes (illness). 

### 2.2. Survey

We created the survey in Qualtrics, a secure online platform for survey hosting. The questionnaire included sociodemographic questions regarding participants’ age, gender (0 = female, 1 = male), time since bereavement (continuous variable); cause of death (1 = natural death, 0 = other), type of relationship (1 = first degree relative, 0 = other), the name of their university/college, followed by two standardized instruments, which were translated according to the four steps described by the World Health Organization: (1) forward translation, (2) expert panel consultation, (3) backward translation and (4) pre-testing of the translated scale [38].

The Adolescent Grief Inventory (AGI) [39] validated for use in populations aged between 12 and 28 years, contains 40 items divided over six factors: sadness, self-blame, anxiety and self-harm, shock, anger and betrayal, and sense of peace (for example: “I had nightmares”). Participants rate the items on a five-point Likert-type scale ranging from “1. Not at all” to “5. Extremely”, indicating how much the statements applied to them during the past month. Item scores are averaged to become the total score. A higher score indicates stronger grief reactions (range 1–5). The Cronbach’s alpha coefficient of the total scale in our sample was 0.90.

The Multidimensional Scale of Perceived Social Support (MSPSS) [40] consists of 12 items divided into 3 subscales: significant other, family, and friends (for example: “I can talk about my problems with my family”). Participants rated each statement on a seven-point Likert-type scale ranging from “1. Very strongly disagree” to “7. Very strongly agree”, based on the past two weeks. Item scores are averaged to become the total scores. Higher scores indicate more perceived social support (range 1–7). The Cronbach’s alpha coefficient of the total scale was 0.91.

The Hogan Grief Reaction Checklist–Panic Behavior (HGRC-PB) [41] includes 14 statements regarding panic and somatic reactions (for example: “I often have headaches”). Participants respond to each item on a five-point Likert-type scale, ranging from “1. Does not describe me very well” to “5. Describes me very well”, regarding how much it applied to them during the past two weeks. Scores are summed and higher scores indicate more physical complaints (range 14–70). The Cronbach’s alpha coefficient in this study was 0.91. 

Finally, the questionnaire asked two open-ended questions: “Have you experienced other somatic complaints in regard to your grief? If so, please specify”. “Have you taken any steps to remedy these somatic complaints? If so, please specify”. 

### 2.3. Data Analysis

We uploaded all data in SPSS v27 [42]. We calculated Pearson/Spearman correlations between the GHRC-PB scores and other somatic complaints scores (HGRC-PB; dummy variable ‘other somatic complaints’) and age, sex, time since bereavement, cause of death, type of relationship with deceased person, and social support scores to check whether we had to control for these variables in the multiple hierarchic regressions. 

We assessed the association between grief (AGI) [39] and somatic complaints (HGRC-PB) [41] with a Pearson correlation. We then performed a multiple hierarchic regression, with the score on the HGRC-PB as a dependent variable, and the significant control variables (step 1), the total AGI score (grief) (step 2) and the interaction term of the standardized total AGI score and the social support score (step 3) as independent variables. The last step was added to investigated whether the relationship between grief and somatic complaints (HGRC-PB) was moderated by social support.

The replies to the additional question “Have you experienced other somatic complaints in regard to your grief? were converted to a dummy variable ‘somatic complaints’ with ‘no’ = 0 and ‘yes’ = 1. We conducted a second hierarchical regression analysis with other somatic complaints as a dependent variable, and the significant control variables (step 1), the total AGI score (grief) and the interaction term of the standardized total AGI score and the social support score (step 3) as independent variables. We also checked the assumptions to perform the regression analyses. The data of the HGRC-PB were not normally distributed, but all other assumptions were met (i.e., linear relationship, no multicollinearity: Variance Inflation Factors (VIF) indices <10 en Tolerance indices >0.10, homoscedasticity).

The replies to the two open-ended questions were analysed descriptively using frequencies and percentages. 

### 2.4. Ethical Approval

At the end of the survey, participants received a link with the contact details of relevant student services at each college and university. They also received the contact details of community counselling centres, and the telephone number of the suicide prevention centre. Throughout the study, participants could contact the researchers in case they had any questions. 

All participants provided online informed consent. The Social and Societal Ethics Committee of the KU Leuven–University of Leuven approved the study (G-2019 10 1762, 19 March 2020).

## 3. Results

The Pearson correlation indicated a significant, positive correlation between reporting somatic complaints (HGRC-PB) and the level of grief (AGI) (*r* = 0.56, *p* < 0.01). The Pearson/Spearman correlations between the variables age, sex, time since bereavement, cause of death, type of relation with deceased person, social support, and the somatic complaints (HGRC-PB, dummy variable) are displayed in Table 1. More somatic complaints (HGRC-PB) were significantly related to a younger age, female gender, no natural cause of death, and less social support. Other somatic complaints (dummy variable) were only significantly related to type of relationship with the deceased person, in that the death of a relative was significantly related with more other somatic symptoms. 

The regression analysis (Table 2) demonstrated that the control variables (age, sex, time since bereavement, cause of death, and social support) explained 11.2% of the somatic complaints (∆*R*^2^ = 0.11, *F*(5, 668) = 16.84, *p* < 0.001). The level of grief explained 23.90% of the somatic complaints, in addition to the control variables (∆*R*^2^ = 0.24, *F*(1, 667) = 246.16, *p* < 0.001), and finally, the interaction between grief and social support did not explain additional variance in somatic complaints (∆*R*^2^ = 0.00, *F*(1, 666) = 0.04, *p =* 0.843), in addition to the control variables and the level of grief. Hence, it was concluded that being female, having less social support and having higher levels of grief all significantly correlated with more somatic complaints (HGRC-PB).

Using a dummy variable ‘somatic complaints’, the Pearson correlation indicated a significant, positive correlation between the experienced complaints and the level of grief (*r* = 0.25, *p* = 0.01). The hierarchic regression analysis (Table 2) showed that the control variable type of relationship with the deceased person was significantly associated with the report of other somatic complaints and explained 4% of the variance of other somatic complaints (∆*R*^2^ = 0.04, *F*(1, 671) = 27.70, *p* < 0.001). The level of grief (AGI) explained 4.45% of the somatic complaints, in addition to the control variable type of relationship with the deceased person (∆*R*^2^ = 0.045, *F*(1, 670) = 33.08, *p* < 0.001), and finally, the interaction between grief and social support did not explain additional variance in other somatic complaints (∆*R*^2^ = 0.001, *F*(1, 669) = 0.75, *p =* 0.387), in addition to the control variable and the level of grief. It was concluded that the type of relationship with the deceased person (first degree relative) and a higher level of grief were positively associated with the dummy variable somatic complaints.

The mean score on the HGRC-PB was 33.15 (*SD* = 12.40), which indicates that the group of participants scored moderate on worries and somatic complaints. Furthermore, 12.94% of the participants (*n* = 89) reported having experienced other somatic complaints (participants could report more than one complaint), including pain/pressure on the chest (*n* = 34), intestine-related complaints such as constipation, indigestion, puking, nausea (*n* = 26), sleep and concentration problems (*n* = 12), muscle strain (*n* = 10), hyperventilation (*n* = 8), and others, such as general illness, skin rash, hair loss, and loss of appetite (*n* = 16). 

A total of 18.60% of all participants (*N* = 688) reported having searched for help to remedy these complaints (HGRC-PB and question about other complaints; Table 3), mostly using medication (*n* = 64).

## 4. Discussion

We found that less social support, no natural cause of death, younger age (for HGRC-PB), type of relationship (for other somatic complaints), and level of grief were positively associated with somatic complaints, and bereaved students reported various complaints such as feeling pain and strains. These results confirmed the hypotheses. The study found that type of relationship (first-degree relatives) was positively associated with somatic complaints. This is in line with the grief literature indicating a stronger impact of bereavement in first-degree relatives compared to other types of relationship [15,43]. The literature also corroborates our finding that the impact of a bereavement is stronger at younger age, while the effect of sex is less clear in the literature [15,43].

The findings are supported by a study by Kaiser and Primavera [31] who reported a correlation between the death of a family member and the development of chronic headache in emerging adulthood, especially when the bereaved person could not process the grief [31]. Moreover, the type of the relationship with the deceased would be associated with reporting somatic complaints, such as headache and breathing problems [4]. Emerging adults who lost a sibling reported more somatic complaints than those who lost a friend. More conflicts in the relationship with the deceased was also associated with reporting more somatic complaints [4]. Further, those who have fewer social contacts may experience overall poorer health compared to those who are socially active [44]. Nonetheless, further research may clarify the role of type of relationship on somatic complaints in bereavement. 

Less than one in five of our participants searched for help regarding their somatic complaints. This is corroborated by research with adult populations indicating that bereaved individuals tend not to seek help even if they experience functional impairments, to the extent that those most vulnerable appear to seek the least help [19,45]. Research with people bereaved by suicide suggested that those bereaved do not readily recognize somatic complaints as grief reactions or reasons for help-seeking [46,47]. Our findings as well as those of the literature are worrisome given the long-term risk of somatic complaints after bereavement. Bolton and colleagues [48] reported elevated rates of chronic illnesses, such as cardiovascular disease, chronic obstructive pulmonary disease, hypertension, and diabetes, in bereaved parents two years after their bereavement, and high levels of traumatic grief at six months have been associated with cancer and heart attacks in widows two years after bereavement [49]. 

Somatic grief reactions may go unrecognized, especially if these occur one or two years after the loss. Somatic reactions may also affect concentration and study motivation, which in turn affects academic achievement [13]. Somatic grief reactions can be more pronounced after a highly distressing loss as through suicide [50]. These observations underscore the crucial importance of exploring somatic grief reactions in bereaved students. 

As bereavement may have long-term health consequences, the literature recommends outreach and early intervention by health professionals. For example, a general practitioner or a practitioner from a student health service can reach out to the bereaved individual or family to assess their health such as cardiac symptoms [51]. Early intervention may also provide an opportunity to initiate subsequent and long-term care [51,52]. Symptoms such as physical pain may occur only after a while, and existing chronic illnesses such as diabetes or respiratory illnesses can exacerbate following bereavement [51]. A general practitioner is also well placed to advise on sleep and healthy lifestyle, including diet and exercise, and initiate mental health treatment or refer to a specialized mental health professional as needed [51]. Usually, general practitioners and student health services are accessible and affordable, and may also offer a listening ear and psychoeducation to the bereaved individuals [51,52]. Usually, student health services have lower thresholds compared to specialized grief or psychological services. Appropriate training of staff at these health services would be a prerequisite for providing adequate support to the bereaved students [53,54,55]. 

The study findings should be considered within certain limitations. This cross-sectional study relied on self-reported data, which may entail a recall bias. Nonetheless, we successfully recruited a large sample with participants from different institutions. As our study was part of a larger project, we could only ask a limited number of questions about somatic complaints. Future studies could use qualitative methods to explore bereaved students’ experiences with somatic complaints and adopt longitudinal designs to measure such complaints over time. The data collection coincided with the onset of the COVID-19 pandemic, and it is unknown whether this affected the participants or the data collection. 

## 5. Conclusions

Our study revealed a positive association between less social support, type of relationship, and the level of grief, and somatic complaints, and bereaved students reported various complaints. Given the potentially devastating impact of bereavement on students’ health and functioning, our findings imply that information and psychoeducation for bereaved students and their social environment must address somatic grief reactions and encourage timely help-seeking for these complaints. In addition, staff members at psychosocial and medical services for students should receive appropriate training to be equipped and skilled to recognize somatic in addition to psychological grief reactions, inquire about such complaints, and provide adequate support to prevent long-term health ramifications. 

## Figures and Tables

**Table 1 ijerph-19-12108-t001:** Pearson (first line)/Spearman (second line) correlations between the control variables and the somatic complaints (HGRC-PB, dummy variable).

	HGRC-PB	Other Somatic Symptoms (Dummy Variable)
Age	−0.11 *	0.05
	−0.12 *	0.04
Sex	−0.25 *	−0.06
	−0.27 *	−0.06
Time since bereavement	−0.03	−0.03
	−0.06	−0.05
Cause of death	−0.13 *	−0.06
	−0.12 *	−0.06
Type of relation	0.07	0.20 *
	0.08	0.20 *
Social Support	−0.16 *	−0.07
	−0.16 *	−0.07

* *p* < 0.01.

**Table 2 ijerph-19-12108-t002:** Final Multiple Hierarchic Regression Analyses of Grief on Somatic Complaints (HGRC-PB).

		Somatic Complaints					
	Variable	*B*	*SE(b)*	*Beta*	*t*	*p*	∆R^2^
Step 1							0.11
	Regression constant	19.51	5.17			<0.001	
	Age	−0.34	0.20	−0.05	−1.70	0.09	
	Sex_female	−5.79	1.11	−0.17	−5.22	<0.001	
	Time since bereavement	−0.01	0.01	−0.03	−0.85	0.40	
	Cause of death	0.90	0.94	0.03	0.96	0.34	
	Social support	−0.10	0.03	−0.11	−3.30	<0.001	
Step 2							0.24
	AGI	11.44	0.74	0.53	15.57	<0.001	
Step 3							0.00
	AGI × Social Support	0.07	0.36	0.01	0.20	0.84	
		Somatic complaints (Dummy)					
	Variable	*B*	*SE(b)*	*Beta*	*t*	*p*	∆R^2^
Step 1							0.04
	Regression constant	−0.21	0.06				
	Type of relationship with deceased person	0.12	0.03	0.15	3.84	<0.001	
Step 2							0.045
	AGI	0.13	0.02	0.22	5.60	<0.001	
Step 3							0.00
	AGI x Social support	−0.01	0.01	−0.03	−0.87		

*B* = standardized regression coefficients. ∆R^2^ = increase in proportion explained variance. AGI: Adolescent Grief Inventory [39], HGRC-PB: Hogan Grief Reaction Checklist–Panic Behavior [41].

**Table 3 ijerph-19-12108-t003:** Frequency Table How Handled Somatic Complaints ^1^ (*N* = 688).

	*n*	Percentage (%)
Medication	64	0.09
Therapy/psychologist	33	0.05
Physiotherapy/osteopathy	19	0.03
Homeopathy/nutrients	16	0.02
Doctor/hospital/operation	10	0.01
Acupuncture/Reiki/Yoga	4	0.01

^1^ Participants could report more than one item.

## Data Availability

According to the ethics approval, data can only be accessed by the members of the research team.

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
