# Peer review of "Association between Grief and Somatic Complaints in Bereaved University and College Students"

_ijerph, 2022, doi:10.3390/ijerph191912108_

Round 1

Reviewer 1 Report

Comment 1 (Abstract and Keywords): The keywords chosen simply repeat most of the variable names already included in the Abstract. Tracking the words of the Abstract so closely will limit the discoverability of the article and its findings in literature searches—which is a key purpose of using keywords.

Comment 2 (Introduction): The Introduction gives a very limited rationale for this particular investigation. As far as I know, an increasing number of studies have identified that bereaved people often suffer from grief and physical and psychological illnesses. However, the manuscript briefly presents “little is known about the impact of grief on somatic complaints in students”, and little justification and rationales are provided to the reader about why it is important to continue to explore such associations in this population.

Comment 3 (Materials and Sampling): It would be useful to demonstrate how to address the missing data, and the appropriateness of the data for standard multiple regression (e.g., normality, etc.).

Comment 4 (Results): As demonstrated in previous research, the time of experiencing bereavement, the dead loved one, and the cause of death significantly influence the bereaved one’s grief and well-being. For example, compared to the death of a friend or sibling, experiencing the death of parents is proved to be more distressful and negatively affects people's health and well-being. Thus, it would be better to add the analysis of the different grief experiences on somatic complaints.  

Comment 5 (Discussion): You mentioned the social support for bereaved students in the Introduction section. The Discussion does not touch on the potential influence of social support in your investigation. It would be better if you could examine the moderating effects of social support on the association between grief and somatic complaints.  

Author Response

Reviewer 1

Comment 1 (Abstract and Keywords):

The keywords chosen simply repeat most of the variable names already included in the Abstract. Tracking the words of the Abstract so closely will limit the discoverability of the article and its findings in literature searches—which is a key purpose of using keywords.

Reply

Thank you for this comment. We have modified the keywords as follows:

Old: bereavement; grief; students; university; somatic symptoms, physical complaints; support; help-seeking

New: bereavement; grief; emerging adults; tertiary education; somatic symptoms, physical complaints; pain; social support; intervention

Comment 2 (Introduction):

The Introduction gives a very limited rationale for this particular investigation. As far as I know, an increasing number of studies have identified that bereaved people often suffer from grief and physical and psychological illnesses. However, the manuscript briefly presents “little is known about the impact of grief on somatic complaints in students”, and little justification and rationales are provided to the reader about why it is important to continue to explore such associations in this population.

Reply

Thank you for this comment. We checked the literature again, and we agree that several studies – in adult populations – report or mention physical or somatic complaints after a bereavement. However, only few studies have been conducted specifically regarding somatic complaints in students. Nonetheless, we have expanded the introduction substantially to strengthen the context of the study. If the reviewer would know other (key) publications that should be references, we would be happy to consider these as well.

Comment 3 (Materials and Sampling):

It would be useful to demonstrate how to address the missing data, and the appropriateness of the data for standard multiple regression (e.g., normality, etc.).

Reply

We have included the following sentences in the section ‘Data analysis’: “We also checked the assumptions to perform the regression analyses. The data of the HGRC-PB was not normally distributed, but all other assumptions were met (i.e., linear relationship, no multicollinearity: Variance Inflation Factors (VIF) indices <10 en Tolerance indices >.10, homoscedasticity).”

Regarding missing data, in the section ‘Study design and sampling’ it was mentioned that incomplete surveys were excluded. There were no other missing data.

Comment 4 (Results):

As demonstrated in previous research, the time of experiencing bereavement, the dead loved one, and the cause of death significantly influence the bereaved one’s grief and well-being. For example, compared to the death of a friend or sibling, experiencing the death of parents is proved to be more distressful and negatively affects people's health and well-being. Thus, it would be better to add the analysis of the different grief experiences on somatic complaints.

Reply

Following the comment, as well as a comment of Reviewer 3, we have now included time since bereavement, type of relationship with the deceased person (first-degree relative, other), and cause of death (natural death, other) in the analysis. The literature indicates that the death of a first-degree relative (in this case, parent and sibling) has a stronger impact than the death of others, and considering the numbers in each category in our study, it was decided to dichotomize type of relationship into first-degree relative vs other). The literature also indicates that violent types of death, such as accidents and suicide, have a stronger impact than natural deaths. Also considering the numbers in each category in our study, we dichotomized cause of death as natural death vs other.

Following a further comment (comment 5), we also included social support and examined the moderating effect of social support on the association between grief and somatic complaints.

We have modified the manuscript accordingly.

Comment 5 (Discussion):

You mentioned the social support for bereaved students in the Introduction section. The Discussion does not touch on the potential influence of social support in your investigation. It would be better if you could examine the moderating effects of social support on the association between grief and somatic complaints.  

Reply

Thank you for the comment. As mentioned above, the analysis now includes the moderating effect of social support on the association between grief and somatic complaints.

Again, many thanks for your valuable feedback, which helped us to improve the manuscript.

Reviewer 2 Report

This is a well-written manuscript, and I commend the authors for producing a high-quality paper.  I wondered why adolescent grief inventory was used since the age range of the participants were between the ages of 18 to 28. This is a minor comment and the authors need to justify the selection of that tool and not another tool that might be more age-appropriate.

Author Response

Reviewer 2

Comment 1

This is a well-written manuscript, and I commend the authors for producing a high-quality paper. I wondered why adolescent grief inventory was used since the age range of the participants were between the ages of 18 to 28. This is a minor comment and the authors need to justify the selection of that tool and not another tool that might be more age-appropriate.

Reply

Thank you for the overall positive evaluation of our manuscript. The Adolescent Grief Inventory was validated in young people aged between 12 and 28 years (Andriessen et al., 2018), which is now mentioned in the methods section (… “validated for use in populations aged between 12 and 28 years”).

Andriessen et al. (2018). The Adolescent Grief Inventory: Development of a novel grief measurement. Journal of Affective Disorders, 240, 203-211.

Reviewer 3 Report

I would like to thank you for giving me this opportunity to evaluate this scientific paper, which focuses on association between grief and somatic complaints in bereaved students

This manuscript reports new findings and is theoretically based on the current literature.

- The manuscript is within the sections’ scope (Wellbeing and Mental Health among Students and Young People)

- This study was well designed, executed, and presented.

- Tables are well presented

- The conclusion is consistent with the evidence presented

- The discussion is relevant

- References are up to date and relevant,

In the discussions section the authors state the following: ‘’Moreover, the type of the relationship with the deceased would be associated with reporting somatic complaints, such as headache and breathing problems [4]. Emerging adults who lost a sibling reported more somatic complaints than those who lost a friend’’

In the Study design page 2 row 95 the authors mention that ‘’About 51% of participants had lost a grandparent, 16% a parent, 5% a sibling, 11% 95 another family member, and 17% a friend or other person. Approximately 75% of the de-96 ceased persons had died through natural causes (illness)’’, I think the paper will be improve if the authors also analyze the impact of who the students have lost and compare the results with the literature.

Author Response

Reviewer 3

Comment 1

I would like to thank you for giving me this opportunity to evaluate this scientific paper, which focuses on association between grief and somatic complaints in bereaved students.

This manuscript reports new findings and is theoretically based on the current literature.

- The manuscript is within the sections’ scope (Wellbeing and Mental Health among Students and Young People)

- This study was well designed, executed, and presented.

- Tables are well presented

- The conclusion is consistent with the evidence presented

- The discussion is relevant

- References are up to date and relevant,

Reply

Thank you very much for the overall positive evaluation of our manuscript.

Comment 2

In the discussions section the authors state the following: ‘’Moreover, the type of the relationship with the deceased would be associated with reporting somatic complaints, such as headache and breathing problems [4]. Emerging adults who lost a sibling reported more somatic complaints than those who lost a friend’’

In the Study design page 2 row 95 the authors mention that ‘’About 51% of participants had lost a grandparent, 16% a parent, 5% a sibling, 11% 95 another family member, and 17% a friend or other person. Approximately 75% of the de-96 ceased persons had died through natural causes (illness)’’, I think the paper will be improve if the authors also analyze the impact of who the students have lost and compare the results with the literature.

Reply

Thank you for this comment. Following this comment, as well as a comment of Reviewer 1, we have now included time since bereavement, type of relationship with the deceased person (first-degree relative, other), and cause of death (natural death, other) in the analysis. Following a further comment (comment 5 of Reviewer 1), we also included social support and examined the moderating effect of social support on the association between grief and somatic complaints.

We have added the following sentences to the Discussion:

The study found that type of relationship (first-degree relatives) was positively associated with somatic complaints. This is in line with the grief literature indicating a stronger impact of bereavement in first-degree relatives compared to other types of relationship [43, 44]. The literature also corroborates our finding that the impact of a bereavement is stronger at younger age, while the effect of sex is less clear in the literature [43, 44].

We also added further implications regarding service delivery, specifically regarding service delivery for bereaved students.

  1. Feigelman, W.; Rosen, Z.; Joiner, T.; Silva, C.; Mueller, A.S. Examining longer-term effects of parental death in adolescents and young adults: Evidence from the national longitudinal survey of adolescent to adult health. Death Stud. 2017, 41, 133–143.
  2. Balk, D.E. Dealing with Dying, Death, and Grief during Adolescence; Routledge: New York, NY, USA, 2014.

We also expanded the discussion with further implications specifically regarding service delivery for bereaved students.

Again, many thanks for your supportive feedback on our manuscript.